# Public perceptions of conflicting information surrounding COVID-19: Results from a nationally representative survey of U.S. adults

Rebekah H. Nagler[1]*, Rachel I. Vogel[2], Sarah E. Gollust[3], Alexander J. Rothman[4], Erika Franklin Fowler[5], Marco C. Yzer[1]

1 Hubbard School of Journalism & Mass Communication, University of Minnesota, Minneapolis, Minnesota, United States of America, 2 Department of Obstetrics, Gynecology & Women's Health, University of Minnesota Medical School, Minneapolis, Minnesota, United States of America, 3 Division of Health Policy and Management, University of Minnesota School of Public Health, Minneapolis, Minnesota, United States of America, 4 Department of Psychology, University of Minnesota, Minneapolis, Minnesota, United States of America, 5 Department of Government, Wesleyan University, Middletown, Connecticut, United States of America

* nagle026@umn.edu

**Data Availability Statement:** All relevant data are available at https://osf.io/x7m98/.

## Abstract

Conflicting information surrounding COVID-19 abounds, from disagreement over the effectiveness of face masks in preventing viral transmission to competing claims about the promise of certain treatments. Despite the potential for conflicting information about COVID-19 to produce adverse public health effects, little is known about whether the U.S. public notices this information, and whether certain population subgroups are particularly likely to do so. To address these questions, we fielded a nationally representative survey of U.S. adults in late April 2020 ($N = 1,007$). Results showed substantial self-reported exposure to conflicting information about COVID-19, with nearly 75% of participants reporting having recently heard such information from health experts, politicians, and/or others. Participants perceived disagreement across a range of COVID-19-related issues, though from politicians more than health experts. Factors including political affiliation, information source use, and personal experience with COVID-19 were associated with perceptions of disagreement. Future research should consider potential cognitive and behavioral consequences of such perceptions.

## Introduction

On February 15, 2020, World Health Organization (WHO) Director-General Tedros Adhanom Ghebreyesus cautioned, "We're not just fighting an epidemic; we're fighting an infodemic" [1]. Ghebreyesus was referring to the proliferation of information, and particularly misinformation, about the novel coronavirus (SARS-CoV-2) and the disease it causes, COVID-19. The rapid spread of inaccurate information and conspiracy theories about COVID-19 via social media and in other spaces poses a clear threat to public understanding and decision making. Yet while much attention to date has been directed toward documenting

**Funding:** This work was supported by a COVID-19 Rapid Response Grant from the Office of the Vice President for Research at the University of Minnesota (OVPR COVID19 #05; PI: RHN). Additional support was provided by a grant from the National Cancer Institute (5R21CA218054-02; PI: RHN). This content is solely the responsibility of the authors and does not necessarily represent the official views of the National Institutes of Health. The funders had no role in study design, data collection and analysis, decision to publish, or preparation of the manuscript. Funder URLs: https://research.umn.edu/funding-awards/ovpr-funding/covid-19-rapid-response-grants and https://cancercontrol.cancer.gov/brp/

**Competing interests:** The authors have declared that no competing interests exist.

and combatting such misinformation [2–5], another aspect of the infodemic has been relatively overlooked: conflicting information surrounding COVID-19. Whereas misinformation refers to false information that is shared either knowingly or accidentally (e.g., wearing a mask can cause deadly rebreathing of exhaled carbon dioxide) [6], conflicting information has been defined as two or more health-related propositions that are logically inconsistent with one another (e.g., wearing a mask does versus does not help prevent viral transmission) [7]. Assessing the extent to which the public notices conflicting information about COVID-19 is critically important, given evidence that exposure to conflicting health messages can translate into adverse public health effects, including confusion about and decreased trust in health recommendations and, in turn, reduced engagement with prevention behaviors [8–10].

Since its outbreak in the United States in early 2020, discourse about COVID-19 has been characterized by substantial disagreement among politicians and health experts alike—observed across a range of issues, including who is most at risk for coronavirus infection, how dangerous such infection is, whether there is adequate access to diagnostic testing, how effective certain treatments are, and how effective personal health and policy-level strategies are in preventing the virus's spread. Such disagreement contributes to the dissemination of conflicting information. Conflicting health information can take many forms, such as inconsistent results across research studies, distinct recommendations among professional organizations, and—perhaps most germane to the COVID-19 context—debate or disagreement about research or recommendations among key stakeholders or sources [11]. A prominent example of such disagreement has been conflicting guidance on face masks [12], but examples have emerged regarding other aspects of the pandemic as well, such as recent and related debate over the prevalence of asymptomatic transmission (i.e., asymptomatic viral spread is common versus rare) and opposing claims about whether drugs such as chloroquine and hydroxychloroquine are effective in treating COVID-19 (i.e., they are effective versus they are ineffective). Although systematic analyses of COVID-19 media coverage are not yet available in the literature, even a cursory glance at coverage to date suggests widespread reporting of conflicting information from sources including the White House [13–15]—a pattern so dominant that it has been summarized in its own right [16]—the World Health Organization [17–19], and other health experts [12, 20, 21].

In general, two major factors can give rise to conflicting health information: the very process of scientific discovery, which features incremental advances and occasional steps backward; and journalistic norms, which emphasize conflict as a core news value [11, 22]. The unique COVID-19 context, characterized by deep scientific uncertainty and severe health consequences, likely amplifies opportunities for conflicting information to arise. Scientists are pursuing questions about a new disease triggered by a novel virus, and they are doing so with great urgency. Given how much is unknown, the rapid evolution of scientific knowledge necessarily increases the likelihood of shifting evidence and, in turn, seemingly ever-changing advice [23, 24]. This picture is complicated by journalistic practice, which not only prioritizes conflict as a news value but also emphasizes novelty and, in turn, has a "default rhythm of constant piecemeal updates [that] is ill-suited to covering an event as large as the pandemic" [23]. This reporting pattern, previously described as the study *du jour* phenomenon [25, 26], could further underscore what the public might perceive to be frequent shifts in recommendations. The sheer volume of news attention to COVID-19—itself a function of the scale and import of the pandemic—could also multiply opportunities for exposure to conflicting information.

Although conditions are ripe for the emergence of conflicting information surrounding COVID-19, little is known about the extent to which the public notices it. Past research gives reason to believe they would: People tend to perceive conflict when such information is prevalent in the media, and this has been observed across health topics as varied as nutrition [8, 27],

mammography screening [28], medications [29], and e-cigarettes [30]. Pew Research Center found initial evidence of public exposure to conflicting information about COVID-19, with 26% of Americans reporting, "I have seen mostly conflicting facts across the sources I turn to for news"; however, this generalized assessment was captured early in the outbreak (March 10–16, 2020) [31]. Observations of media coverage since then point to greater opportunities for exposure to conflict, but public perceptions of conflicting information surrounding COVID-19 need to be assessed systematically. Also unknown is whether certain population subgroups are particularly likely to notice such information. Although few studies have examined correlates of conflicting health information exposure [27, 29], factors such as sociodemographic characteristics, personal experience with COVID-19, geographic context, and use of COVID-19 information sources could shape public perceptions. For example, if someone has had personal experience with COVID-19 or lives in an area particularly hard hit by the disease, this could make it more personally salient, which, in turn, could heighten their attention to COVID-19 information and increase the likelihood that they will notice conflict and disagreement. Documenting these associations will enable us to identify subpopulations who could be particularly susceptible to conflicting information and its potential downstream consequences.

Moreover, it is important to examine whether the public perceives disagreement among health experts, politicians, or both—a necessary differentiation, given the extent to which COVID-19 has been politicized [32]. The politicization of health issues, defined as when political cues become integrated into those issues' public presentation (e.g., when politicians' perspectives appear in news media coverage of an issue to either endorse or highlight political conflict), has been well documented in recent years—among issues as wide-ranging as the human papillomavirus (HPV) vaccine, mammography screening, and the Affordable Care Act (ACA)) [33, 34]. In the context of COVID-19 in the U.S., political cues have been present from the outset, with rhetoric from not only the White House but also governors and other politicians; such discourse has occurred alongside COVID-19-related messaging from health experts, including federal, state, and local health department officials and scientists at academic research institutions [32]. Ultimately, then, there could be opportunities for the public to perceive disagreement among both politicians and health experts, and the level of perceived disagreement could vary across these sources. Assessing these possibilities is critical, as the source of conflicting information could influence whether people notice it and how they respond to it.

The current study addresses two overarching research questions: 1) to what extent does the U.S. public perceive conflicting information surrounding COVID-19, whether from health experts, politicians, or both; and 2) what factors are associated with these perceptions? To answer these questions, we draw on data from a nationally representative survey of U.S. adults conducted in late April 2020, in which participants reported their overall exposure to conflicting information surrounding COVID-19, as well as their perceptions of debate or disagreement among health experts and politicians across a set of specific issues. To better understand observed perceptions of disagreement, we examine several potential correlates of these perceptions, including sociodemographic characteristics, personal experience with COVID-19, geographic context, and use of COVID-19 information sources.

## Materials and methods

### Sample and procedure

Data reported here were collected as part of the AmeriSpeak Omnibus Survey, fielded from April 23–27, 2020 (*N* = 1,007). The Omnibus is a multi-client shared cost survey that is conducted bi-weekly among a nationally representative sample of ~1,000 U.S. adults aged 18 or

older by NORC at the University of Chicago. Omnibus participants are drawn from NORC's AmeriSpeak Panel, a probability-based panel of approximately 43,000 households designed to be representative of the U.S. household population. To recruit panel members, NORC randomly selects U.S. households using area probability and address-based sampling; sampled households are then contacted via mail, telephone, and face-to-face field interviews [35]. The panel provides sample coverage of approximately 97% of the U.S. household population; those excluded include those with P.O. Box only addresses, some addresses not listed in the USPS Delivery Sequence File, and some newly constructed dwellings. The Omnibus survey is administered in English in mixed mode, with approximately 85% of the interviews conducted online and 15% by phone. On average, AmeriSpeak panelists participate in 2–4 surveys per month; to minimize respondent burden, NORC limits panelist participation to 4 surveys per month.

Our team added several survey questions to the late April 2020 Omnibus instrument, a subset of which are included in the current study. Data not analyzed here come from questions that assessed participants' perceptions of disparities in COVID-19 mortality, other COVID-19-related cognitions (e.g., self-efficacy to reduce risk of infection), patterns of information avoidance, and past mitigation behaviors (e.g., stockpiling groceries and other supplies). Data that describe levels of public awareness of disparities in COVID-19 mortality and correlates of that awareness are reported elsewhere [36]. The late April 2020 Omnibus instrument had a total average completion time of 20 minutes.

## Ethics statement

The University of Minnesota Institutional Review Board approved this study (STUDY00009529), determining it to have met the criteria for exemption (Category 2). All participants had previously been consented by NORC to participate in the AmeriSpeak Omnibus Survey.

## Measures

**Perceptions of conflicting information surrounding COVID-19.** Perceptions of conflicting information surrounding COVID-19 were assessed in two ways. To assess *overall self-reported exposure to conflicting information*, we adapted a previously validated measure of conflicting health information exposure to the COVID-19 context [27]. Participants were asked, "Thinking now about the past few weeks, how much conflicting or contradictory information have you heard about COVID-19 (coronavirus), whether from health experts, politicians, and/ or others?" Response options included "None" (1), "A little" (2), "Some" (3), and "A lot" (4). This measure captures a generalized assessment of cross-source exposure to conflicting information about COVID-19, but it does not provide a more nuanced, source- and issue-specific examination. We therefore asked a series of questions to assess *public perceptions of debate or disagreement surrounding COVID-19*—the type of conflicting information most relevant to the COVID-19 context—prior to the more generalized assessment, so that participants would not be primed to think about conflicting information when responding to these more nuanced questions.

One set of questions focused on health experts, and an identical set of questions focused on politicians. For the health expert questions, we provided the following introduction, adapted from past research on public perceptions of politicized health controversies [37]: "Some health issues seem to arouse a lot of debate or disagreements among health experts, while there is more agreement on other health issues." This was followed by defining language: "When we say health experts, we mean, for example, scientists at the Centers for Disease Control (CDC), the National Institutes of Health (NIH), such as Dr. Anthony Fauci, state or local health

departments, and academic research institutions." Then participants were asked, "Based on what you've read, seen or heard in the past week, how much disagreement do you think there is about the following aspects of COVID-19 (coronavirus) among health experts: i) Who is most at risk of being infected with COVID-19 (coronavirus); ii) How dangerous it is for someone to become infected with COVID-19 (coronavirus); iii) Whether there is adequate access to testing for COVID-19 (coronavirus); and iv) Whether the drugs chloroquine and hydroxy-chloroquine are effective in treating COVID-19 (coronavirus)." For each aspect, response options included "No disagreement" (1), "A little disagreement" (2), "Some disagreement" (3), and "A lot of disagreement" (4). Again referencing what they read, saw, or heard in the past week, participants were asked, "How much disagreement do you think there is among health experts about the effectiveness of the following strategies for preventing the spread of COVID-19 (coronavirus): i) Keeping 6 feet away from other people, except those you live with; ii) Wearing a mask or other face covering when out in public; iii) Keeping schools closed; iv) Keeping all businesses closed except those considered essential (e.g., grocery stores, pharmacies); v) Self-quarantining when sick; and vi) Washing your hands with soap several times per day." For each strategy, response options again included "No disagreement" (1), "A little disagreement" (2), "Some disagreement" (3), and "A lot of disagreement" (4). The same sets of questions were asked about politicians, with similar introductory and defining language: "Some health issues seem to arouse a lot of debate or disagreements among politicians, while there is more agreement on other health issues. When we say politicians, we mean, for example, the president and U.S. Congress, governors and state legislators, and local mayors and city council members." Health expert and politician question blocks appeared in random order, such that some participants saw the two health expert batteries first, and some saw the two politician batteries first. Within each battery, the order in which specific aspects or strategies appeared was randomized as well.

The four aspects and six strategies were purposively selected from the universe of COVID-19-related issues prevalent in the media during mid/late April, with the goal of including a range of issues about which perceptions of disagreement might vary. Because our primary research question was concerned with public perceptions of conflicting information surrounding COVID-19 rather than issue-specific perceptions, we averaged across items to generate four summary indices, which are the focus of our analyses: perceptions of disagreement among 1) health experts and 2) politicians about specific aspects of COVID-19, and perceptions of disagreement among 3) health experts and 4) politicians about the effectiveness of strategies for preventing the spread of COVID-19. Issue-specific perceptions are reported in an appendix (S1 and S2 Tables). For both the health expert and politician question blocks, we kept the aspects and strategies indices separate because of the conceptual distinctions between them. All six items in the strategies index assess the effectiveness of strategies for preventing the spread of COVID-19; the four items in the aspects index, though more loosely connected to one another, are concerned with perceptions of risk, testing, and treatment. The indices are not so strongly correlated so as to suggest they are capturing the same phenomena (aspects and strategies indices, health experts: $r = 0.55$; politicians: $r = 0.55$).

**Correlates of perceptions of disagreement surrounding COVID-19.** Potential correlates of perceived disagreement fell into several categories: sociodemographic characteristics, personal experience with COVID-19, geographic context, and use of COVID-19 information sources.

*Sociodemographic characteristics.* NORC provides demographic profile data as part of their standard data delivery, including gender (male, female), age (18–29, 30–44, 45–59, 60+), race/ethnicity (White, non-Hispanic; Black, non-Hispanic; Hispanic; other), education (less than high school, high school graduate or equivalent, some college, bachelor's degree or above), and household income (<$25,000, $25,000-$49,999, $50,000-$74,999, $75,000-$99,999, $100,000+).

We assessed political affiliation using a 7-point self-placement measure [38]. Participants were asked, "Generally speaking, would you call yourself. . .", with response options "A strong Democrat" (1), "A Democrat" (2), "Someone who leans Democratic" (3), "An Independent" (4), "Someone who leans Republican" (5), "A Republican" (6), and "A strong Republican" (7). These options were subsequently collapsed into "Democrat," "Independent," and "Republican."

*Personal experience with COVID-19.* Participants were asked two questions to gauge their personal experience with COVID-19. First, they were asked, "Have you been told by a doctor or other health care provider that you have COVID-19 (coronavirus)?" Response options included "No" (1), "No, but I have or have had concerning symptoms" (2), "Yes," (3), and "I don't know" (4). Then they were asked, "Do you personally know anyone, other than yourself, who has been told by a doctor or other health care provider that they have COVID-19 (coronavirus)?" Response options included "No" (1), "Yes," (2), and "I don't know" (3). The two items were combined to create a summary measure of personal experience with COVID-19 (yes to either/have or have had concerning symptoms (1), no/I don't know (0)).

*Geographic context.* Geographic context was assessed in two ways. First, NORC provided participant profile data on geographic region as part of their standard data delivery (Northeast, Midwest, South, West). Second, we merged county-level COVID-19 mortality rate data from Kaiser Health News (case counts as of April 22, 2020) [39] with our survey data, which we then categorized by quartile (<1 per 100,000, 1–3 per 100,000, 3–9 per 100,000, >9 per 100,000).

*Use of COVID-19 information sources.* To assess participants' COVID-19 information use patterns, we asked, "Thinking now about specific information sources, which of the following sources have you turned to for information about COVID-19 (coronavirus) in the past week: i) Fox News or its website; ii) MSNBC or its website; iii) CNN or its website; iv) NPR or its website; v) The New York Times or its website; vi) The Washington Post or its website; vii) Local television news in your area or their websites; viii) Local newspaper in your area or its website; ix) National network news (ABC World News Tonight, CBS Evening News, or NBC Nightly News) or their websites; x) White House press briefings; xi) State governor briefings; xii) Centers for Disease Control (CDC); xiii) World Health Organization (WHO); xiv) State or local health department; xv) Other people (such as family, friends, or co-workers); and xvi) Another source (specify)." Participants were asked to select all that apply; the order in which sources appeared was randomized. Participants selected 4.2 sources on average (*SD* = 2.7); only 22 participants selected the "another source" option without selecting at least one other source. Guided by the Pew Research Center's public opinion work on news and information sources, both in the COVID-19 context and more generally [40, 41], specific sources were collapsed to generate broader information source categories: cable news (sources i-iii); national news (iv-vi, ix); local news (vii-viii); direct political sources (x-xi); direct health sources (xii-xiv); and interpersonal sources (xv) (yes (1), no (0) for each source category).

In addition, to gauge the extent to which participants actively looked for information, we asked, "In the past week, how often have you checked for news about COVID-19 (coronavirus) from any source?" Response options included "Every hour or more frequently" (1), "About 5–6 times a day" (2), "About 2–3 times a day" (3), "Once a day" (4), "Multiple times per week, but less than once a day" (5), "Less than once per week," (6), and "Never" (7). These were subsequently collapsed into three categories: "Two or more times a day," "Once a day," and "Less than once a day."

### Analytic approach

To assess the prevalence of public perceptions of conflicting information surrounding COVID-19, frequency analyses were conducted for the generalized assessment of self-reported exposure to conflicting information, and descriptive statistics were calculated for the four

summary source-specific indices of perceptions of disagreement. Multivariable linear regression models were estimated to predict each of these four summary indices: Perceptions of disagreement among 1) health experts and 2) politicians about specific aspects of COVID-19, and among 3) health experts and 4) politicians about the effectiveness of strategies for preventing the spread of COVID-19. Models included the following independent variables, as defined above: gender, age, race/ethnicity, education, household income, political affiliation, personal experience with COVID-19, region, county-level COVID-19 mortality rate, COVID-19 information sources, and frequency of checking news about COVID-19. These variables were assessed for potential collinearity using both Spearman's correlation and variance inflation factors (VIF); there was no evidence of collinearity. As a sensitivity analysis, models were conducted to account for the potential correlation among multiple participants within county; because few participants were from a given county, and no significant differences in conclusions were observed, these models are not presented. NORC survey weights—based on national census benchmarks and balanced by gender, age, education, race/ethnicity, and region—were applied to adjust for potential biases in sampling and nonresponse to produce nationally representative estimates. Across models, *P*-values of < .05 were considered statistically significant. All analyses were conducted in SAS version 9.4 (Cary, NC).

## Results

### Participant characteristics

Just over one-third of participants (35.2%) reported having had personal experience with COVID-19. About 40% of participants (42.8%) were Democrats; 29.8% were Republicans, and 27.4% were Independents. The information sources that participants reported turning to most for information about COVID-19 were local news (55.6%), direct political sources (53.2%), cable news (51.6%), and direct health sources (46.6%). Nearly half of participants (48.9%) reported checking for news about COVID-19 two or more times a day. Additional participant characteristics are reported in Table 1.

### Perceptions of conflicting information surrounding COVID-19

Overall, nearly three-quarters of participants (72.3%) reported hearing some or a lot of conflicting information about COVID-19, whether from health experts, politicians, and/or others; only 3.3% reported no exposure to such information. Consistent with this pattern, participants reported perceiving debate or disagreement about specific aspects of COVID-19 among both health experts ($M$ = 2.28, 95% CI: 2.21–2.34, range = 1–4, Cronbach's $\alpha$ = 0.79) and politicians ($M$ = 2.68, 95% CI: 2.61–2.74, range = 1–4, Cronbach's $\alpha$ = 0.80), where 2 was "a little disagreement" and 3 was "some disagreement." They also reported perceiving disagreement about the effectiveness of strategies for preventing viral spread, although to a lesser extent—again among health experts ($M$ = 1.61, 95% CI: 1.55–1.66, range = 1–4, Cronbach's $\alpha$ = 0.84) and politicians ($M$ = 1.97, 95% CI: 1.91–2.03, range = 1–4, Cronbach's $\alpha$ = 0.85). Although disagreement was observed across these two sources, participants perceived even greater disagreement among politicians, as evident in the non-overlapping confidence intervals for both the aspects and strategies indices (Tables 2 and 3, respectively; both $p$ < .001) and issue-specific perceptions (S1 and S2 Tables).

### Correlates of perceptions of disagreement surrounding COVID-19

Several factors were significantly associated with perceptions of disagreement about specific aspects of COVID-19 (Table 2). Participants who reported personal experience with COVID-

**Table 1. Sample characteristics (*N* = 1,007).**

| Variable | Weighted %[a] |
|---|---|
| **Gender** | |
| Male | 48.6 |
| Female | 51.4 |
| **Age (years)** | |
| 18–29 | 18.1 |
| 30–44 | 26.7 |
| 45–59 | 24.5 |
| 60+ | 30.7 |
| **Race/ethnicity** | |
| White, non-Hispanic | 62.6 |
| Black, non-Hispanic | 12.0 |
| Hispanic | 16.5 |
| Other | 8.9 |
| **Education** | |
| Less than high school | 8.8 |
| High school graduate or equivalent | 27.5 |
| Some college | 28.5 |
| Bachelor's degree or above | 35.3 |
| **Household income** | |
| <$25,000 | 20.5 |
| $25,000-$49,999 | 25.6 |
| $50,000-$74,999 | 18.5 |
| $75,000-$99,999 | 12.8 |
| $100,000+ | 22.5 |
| **Political affiliation** | |
| Republican | 29.8 |
| Independent | 27.4 |
| Democrat | 42.8 |
| **Personal experience with COVID-19** | |
| Yes | 35.2 |
| No | 64.8 |
| **Region** | |
| Northeast | 17.6 |
| Midwest | 20.7 |
| South | 37.8 |
| West | 23.9 |
| **County-level COVID-19 mortality rate** | |
| <1 per 100,000 | 23.9 |
| 1–3 per 100,000 | 25.3 |
| 3–9 per 100,000 | 26.4 |
| >9 per 100,000 | 24.4 |
| **COVID-19 information sources[b]** | |
| Cable news | 51.6 |
| National news | 24.5 |
| Local news | 55.6 |
| Direct political sources | 53.2 |
| Direct health sources | 46.6 |

(*Continued*)

**Table 1.**  (Continued)

| Variable | Weighted %[a] |
|---|---|
| Interpersonal sources | 23.3 |
| **How often check news about COVID-19** | |
| Two or more times a day | 48.9 |
| Once a day | 28.5 |
| Less than once a day | 22.6 |

[a] Percentages may not sum to 100 due to missing data and rounding.

[b] Participants could check all that apply.

19 perceived greater disagreement among both health experts ($p$ = .020) and politicians ($p$ = .045) than did those without personal experience. Political affiliation also was significantly associated with perceptions of disagreement, but only for disagreement among politicians ($p$ = .001): both Democrats and Independents tended to perceive greater disagreement among politicians than did Republicans. Other sociodemographic characteristics associated with perceived disagreement included race/ethnicity and education. Hispanic participants tended to perceive less disagreement among politicians ($p$ = .035), compared with non-Hispanic White participants, and participants with a Bachelor's degree or above reported perceiving more disagreement among politicians ($p$ = .010) than those with less than a high school education. Several information sources were associated with perceptions of disagreement as well. Specifically, participants attending to national news reported perceiving greater disagreement among politicians ($p$ = .043) and, to some extent, less disagreement among health experts ($p$ = .061) than those who did not use this source to obtain information about COVID-19. Those attending to direct health sources reported perceiving greater disagreement among politicians ($p$ = .015).

There were also several factors significantly associated with perceptions of disagreement about the effectiveness of strategies for preventing the spread of COVID-19 (Table 3). Again participants with personal experience with COVID-19 reported perceiving greater disagreement, though this association was statistically significant only for disagreement among politicians ($p$ = .030). Compared to the aspects findings, political affiliation was not significantly associated with perceptions of disagreement among either politicians ($p$ = .688) or health experts ($p$ = .096). Race/ethnicity was again associated with perceived disagreement, with Hispanic participants tending to perceive less disagreement among politicians ($p$ = .003), compared with their non-White Hispanic counterparts. In addition, older participants reported perceiving less disagreement than younger participants, and this was observed among both health experts ($p$ = .016) and politicians ($p$ = .020). Consistent with the aspects results, those attending to national news reported perceiving greater disagreement among politicians ($p$ = .021) and, to some extent, less disagreement among health experts ($p$ = .058). Participants attending to direct political sources reported perceiving less disagreement among both health experts ($p$ = .031) and politicians ($p$ = .028), whereas those attending to direct health sources reported perceiving greater disagreement among politicians ($p$ = .012).

## Discussion

The primary goal of this study was to assess the extent to which the U.S. public perceives conflicting information surrounding COVID-19. This dimension of the COVID-19 infodemic has been not been assessed systematically, despite widespread media coverage of conflicting information about COVID-19 and the potential for such content to produce adverse affective, cognitive, and behavioral responses. To address this research question, we fielded a nationally

**Table 2. Multivariable linear regression models predicting perceptions of disagreement among health experts and politicians about aspects of COVID-19 (coronavirus).**

| | Among health experts | | | Among politicians | | |
|---|---|---|---|---|---|---|
| | N | M (SE) | 95% CI | N | M (SE) | 95% CI |
| Summary index (range = 1–4) | 996 | 2.28 (0.03) | 2.21, 2.34 | 997 | 2.68 (0.03) | 2.61, 2.74 |
| | Among health experts (N = 977) | | | Among politicians (N = 977) | | |
| Variable | b | SE | p | b | SE | p |
| Gender | | | 0.528 | | | 0.696 |
| Female (ref) | 0.00 | | | 0.00 | | |
| Male | 0.04 | 0.07 | | -0.03 | 0.07 | |
| Age (years) | | | 0.058 | | | 0.081 |
| 18–29 (ref) | 0.00 | | | 0.00 | | |
| 30–44 | -0.18 | 0.11 | | 0.01 | 0.12 | |
| 45–59 | -0.26 | 0.12 | | 0.05 | 0.12 | |
| 60+ | -0.06 | 0.12 | | 0.21 | 0.12 | |
| Race/ethnicity | | | 0.806 | | | 0.035 |
| White, non-Hispanic (ref) | 0.00 | | | 0.00 | | |
| Black, non-Hispanic | 0.08 | 0.11 | | -0.07 | 0.12 | |
| Hispanic | -0.06 | 0.12 | | -0.34 | 0.12 | |
| Other | 0.03 | 0.12 | | -0.01 | 0.12 | |
| Education | | | 0.746 | | | 0.010 |
| Less than high school (ref) | 0.00 | | | 0.00 | | |
| High school graduate or equivalent | -0.07 | 0.18 | | -0.14 | 0.15 | |
| Some college | -0.04 | 0.17 | | -0.05 | 0.14 | |
| Bachelor's degree or above | -0.11 | 0.17 | | 0.14 | 0.14 | |
| Household income | | | 0.247 | | | 0.822 |
| <$25,000 (ref) | 0.00 | | | 0.00 | | |
| $25,000-$49,999 | -0.26 | 0.11 | | -0.12 | 0.12 | |
| $50,000-$74,999 | -0.17 | 0.12 | | -0.06 | 0.12 | |
| $75,000-$99,999 | -0.22 | 0.12 | | -0.07 | 0.12 | |
| $100,000+ | -0.20 | 0.11 | | -0.04 | 0.12 | |
| Political affiliation | | | 0.437 | | | 0.001 |
| Republican (ref) | 0.00 | | | 0.00 | | |
| Independent | -0.03 | 0.09 | | 0.28 | 0.09 | |
| Democrat | -0.11 | 0.09 | | 0.31 | 0.09 | |
| Personal experience with COVID-19 | | | 0.020 | | | 0.045 |
| No (ref) | 0.00 | | | 0.00 | | |
| Yes | 0.16 | 0.07 | | 0.14 | 0.07 | |
| Region | | | 0.125 | | | 0.653 |
| Northeast (ref) | 0.00 | | | 0.00 | | |
| Midwest | 0.10 | 0.11 | | 0.12 | 0.10 | |
| South | 0.17 | 0.11 | | 0.05 | 0.11 | |
| West | 0.27 | 0.12 | | 0.09 | 0.11 | |
| County-level COVID-19 mortality rate | | | 0.757 | | | 0.606 |
| <1 per 100,000 (ref) | 0.00 | | | 0.00 | | |
| 1–3 per 100,000 | 0.02 | 0.09 | | 0.05 | 0.09 | |
| 3–9 per 100,000 | -0.06 | 0.09 | | 0.08 | 0.09 | |
| >9 per 100,000 | 0.02 | 0.11 | | -0.04 | 0.10 | |
| COVID-19 information sources | | | | | | |

*(Continued)*

**Table 2.** (Continued)

| | Among health experts | | | Among politicians | | |
|---|---|---|---|---|---|---|
| | *N* | *M (SE)* | 95% CI | *N* | *M (SE)* | 95% CI |
| **Summary index (range = 1–4)** | **996** | **2.28 (0.03)** | **2.21, 2.34** | **997** | **2.68 (0.03)** | **2.61, 2.74** |
| | Among health experts (*N* = 977) | | | Among politicians (*N* = 977) | | |
| **Variable** | *b* | *SE* | *p* | *b* | *SE* | *p* |
| Cable news | | | 0.526 | | | 0.269 |
| No (ref) | 0.00 | | | 0.00 | | |
| Yes | 0.04 | 0.07 | | 0.08 | 0.07 | |
| National news | | | 0.061 | | | 0.043 |
| No (ref) | 0.00 | | | 0.00 | | |
| Yes | -0.16 | 0.09 | | 0.16 | 0.08 | |
| Local news | | | 0.524 | | | 0.364 |
| No (ref) | 0.00 | | | 0.00 | | |
| Yes | 0.04 | 0.07 | | 0.06 | 0.07 | |
| Direct political sources | | | 0.090 | | | 0.086 |
| No (ref) | 0.00 | | | 0.00 | | |
| Yes | -0.12 | 0.07 | | -0.12 | 0.07 | |
| Direct health sources | | | 0.103 | | | 0.015 |
| No (ref) | 0.00 | | | 0.00 | | |
| Yes | 0.12 | 0.07 | | 0.18 | 0.07 | |
| Interpersonal sources | | | 0.516 | | | 0.458 |
| No (ref) | 0.00 | | | 0.00 | | |
| Yes | 0.05 | 0.08 | | 0.06 | 0.07 | |
| **How often check news about COVID-19** | | | 0.190 | | | 0.774 |
| Less than once a day (ref) | 0.00 | | | 0.00 | | |
| Once a day | -0.07 | 0.10 | | -0.07 | 0.10 | |
| Two or more times a day | -0.16 | 0.10 | | -0.04 | 0.10 | |

representative survey of ~1,000 U.S. adults in late April 2020 and found substantial self-reported exposure to conflicting information about COVID-19, with nearly 75% of participants reporting that in recent weeks they had heard such information from health experts, politicians, and/or others. In addition, results showed that participants perceived disagreement across a range of COVID-19-related issues, among both health experts and politicians, suggesting that such exposure is not confined to particular issues or sources and instead may be cumulative. That said, there were systematic differences in perceived disagreement across sources—overall, participants noticed more disagreement among politicians than health experts—and mean levels of perceived disagreement tended to be higher for specific aspects of COVID-19 than the effectiveness of strategies for preventing its spread.

These descriptive patterns may be better understood by examining factors associated with perceived disagreement surrounding COVID-19. For example, there was at least some evidence that perceptions could be shaped by one's political affiliation. For specific aspects of COVID-19, Democrats and Independents perceived greater disagreement among politicians than did Republicans. This pattern could reflect motivated reasoning [42]: If the White House is what came to mind when Democrats and Independents read "politicians," then that cue could have prompted them to perceive and ultimately interpret disagreement in ways consistent with their political beliefs (e.g., the president routinely contradicts himself, does not tell the truth, and so forth), whereas the same cue could have triggered the opposite response

**Table 3. Multivariable linear regression models predicting perceptions of disagreement among health experts and politicians about the effectiveness of strategies for preventing the spread of COVID-19 (coronavirus).**

| | Among health experts | | | Among politicians | | |
|---|---|---|---|---|---|---|
| | N | M (SE) | 95% CI | N | M (SE) | 95% CI |
| Summary index (range = 1–4) | 1002 | 1.61 (0.03) | 1.55, 1.66 | 1000 | 1.97 (0.03) | 1.91, 2.03 |
| | Among health experts (N = 981) | | | Among politicians (N = 980) | | |
| Variable | b | SE | p | b | SE | p |
| **Gender** | | | 0.393 | | | 0.816 |
| Female (ref) | 0.00 | | | 0.00 | | |
| Male | -0.05 | 0.05 | | -0.02 | 0.07 | |
| **Age (years)** | | | 0.016 | | | 0.020 |
| 18–29 (ref) | 0.00 | | | 0.00 | | |
| 30–44 | -0.23 | 0.10 | | -0.31 | 0.12 | |
| 45–59 | -0.31 | 0.10 | | -0.37 | 0.12 | |
| 60+ | -0.27 | 0.10 | | -0.33 | 0.12 | |
| **Race/ethnicity** | | | 0.928 | | | 0.003 |
| White, non-Hispanic (ref) | 0.00 | | | 0.00 | | |
| Black, non-Hispanic | 0.03 | 0.10 | | -0.07 | 0.12 | |
| Hispanic | -0.03 | 0.10 | | -0.33 | 0.09 | |
| Other | 0.03 | 0.08 | | -0.20 | 0.11 | |
| **Education** | | | 0.076 | | | 0.788 |
| Less than high school (ref) | 0.00 | | | 0.00 | | |
| High school graduate or equivalent | 0.15 | 0.14 | | 0.06 | 0.14 | |
| Some college | -0.06 | 0.13 | | 0.04 | 0.13 | |
| Bachelor's degree or above | -0.03 | 0.13 | | 0.10 | 0.13 | |
| **Household income** | | | 0.461 | | | 0.253 |
| <$25,000 (ref) | 0.00 | | | 0.00 | | |
| $25,000-$49,999 | -0.15 | 0.09 | | -0.15 | 0.010 | |
| $50,000-$74,999 | -0.15 | 0.10 | | 0.03 | 0.11 | |
| $75,000-$99,999 | -0.09 | 0.11 | | -0.01 | 0.11 | |
| $100,000+ | -0.14 | 0.09 | | -0.02 | 0.11 | |
| **Political affiliation** | | | 0.096 | | | 0.688 |
| Republican (ref) | 0.00 | | | 0.00 | | |
| Independent | -0.12 | 0.08 | | 0.06 | 0.08 | |
| Democrat | -0.16 | 0.07 | | 0.07 | 0.09 | |
| **Personal experience with COVID-19** | | | 0.090 | | | 0.030 |
| No (ref) | 0.00 | | | 0.00 | | |
| Yes | 0.10 | 0.06 | | 0.14 | 0.07 | |
| **Region** | | | 0.117 | | | 0.943 |
| Northeast (ref) | 0.00 | | | 0.00 | | |
| Midwest | 0.12 | 0.09 | | -0.01 | 0.10 | |
| South | 0.13 | 0.09 | | -0.02 | 0.10 | |
| West | 0.21 | 0.09 | | 0.03 | 0.10 | |
| **County-level COVID-19 mortality rate** | | | 0.329 | | | 0.791 |
| <1 per 100,000 (ref) | 0.00 | | | 0.00 | | |
| 1–3 per 100,000 | -0.14 | 0.08 | | -0.06 | 0.08 | |
| 3–9 per 100,000 | -0.07 | 0.07 | | 0.02 | 0.08 | |
| >9 per 100,000 | -0.07 | 0.09 | | -0.04 | 0.09 | |
| **COVID-19 information sources** | | | | | | |

*(Continued)*

**Table 3.** (Continued)

| | Among health experts | | | Among politicians | | |
|---|---|---|---|---|---|---|
| | N | M (SE) | 95% CI | N | M (SE) | 95% CI |
| **Summary index (range = 1–4)** | 1002 | 1.61 (0.03) | 1.55, 1.66 | 1000 | 1.97 (0.03) | 1.91, 2.03 |
| | Among health experts (N = 981) | | | Among politicians (N = 980) | | |
| **Variable** | b | SE | p | b | SE | p |
| Cable news | | | 0.141 | | | 0.312 |
| No (ref) | 0.00 | | | 0.00 | | |
| Yes | 0.09 | 0.06 | | 0.06 | 0.06 | |
| National news | | | 0.058 | | | 0.021 |
| No (ref) | 0.00 | | | 0.00 | | |
| Yes | -0.13 | 0.07 | | 0.18 | 0.08 | |
| Local news | | | 0.727 | | | 0.702 |
| No (ref) | 0.00 | | | 0.00 | | |
| Yes | 0.02 | 0.05 | | -0.02 | 0.06 | |
| Direct political sources | | | 0.031 | | | 0.028 |
| No (ref) | 0.00 | | | 0.00 | | |
| Yes | -0.13 | 0.06 | | -0.14 | 0.07 | |
| Direct health sources | | | 0.200 | | | 0.012 |
| No (ref) | 0.00 | | | 0.00 | | |
| Yes | 0.08 | 0.06 | | 0.16 | 0.06 | |
| Interpersonal sources | | | 0.799 | | | 0.785 |
| No (ref) | 0.00 | | | 0.00 | | |
| Yes | -0.02 | 0.07 | | -0.02 | 0.07 | |
| **How often check news about COVID-19** | | | 0.217 | | | 0.202 |
| Less than once a day (ref) | 0.00 | | | 0.00 | | |
| Once a day | -0.06 | 0.08 | | -0.12 | 0.09 | |
| Two or more times a day | -0.13 | 0.08 | | -0.17 | 0.09 | |

among Republicans (e.g., the White House is a trustworthy source). Future research should investigate these possibilities, given scholarly concern about motivated reasoning in the COVID-19 context [32].

Additionally, where and how people obtain their information about COVID-19 might shape their perceptions of disagreement. For instance, across aspects and strategies, participants who reported turning to direct health sources (international/national health organizations like WHO and CDC, state and local health departments) tended to perceive more disagreement among politicians. In contrast, those attending to direct political sources (White House and/or governor briefings) tended to perceive less disagreement among both health experts and politicians. We can only speculate as to why, but it could be that there is more unified messaging delivered via government briefings—which, at the federal and state levels, can feature both health experts and politicians. By comparison, direct health sources, such as health department and organization officials, might more readily acknowledge scientific uncertainty and the concomitant shifts in guidance, yet the public might attribute such shifts in guidance to political sources involved in such communications (e.g., state government officials working for the health department). Further, there was evidence that participants who reported turning to national news perceived greater disagreement among politicians and, to some extent, less disagreement among health experts. A cursory review of national news headlines suggests that both politicians and health experts have issued contradictory messages surrounding COVID-19 [12, 13, 17], but the public might perceive this to be particularly true of politicians.

Results also showed that one's personal experience with COVID-19 was associated with perceptions of conflict, whereas the broader geographic context in which one lives seemed to exert less influence. Personal experience might have made the disease more salient, perhaps encouraging participants to pay greater attention and, in turn, increase the likelihood that they notice more disagreement—especially about issues such as how dangerous it is and possible treatments. Where someone lives, and whether there was substantial COVID-19 mortality in their area, did not seem to be consequential here, at least in a multivariable model that took people's personal experience with the disease into account. There was also evidence of associations with certain demographics beyond political affiliation, such as Hispanic ethnicity and age, though it will be important to see whether such patterns hold up in future studies.

This study has several strengths, including the use of population-based data and the adaptation of survey measures previously developed to assess self-reported exposure to conflicting health information [27] and perceptions of politicized health controversies [37]. Nonetheless, results should be considered with several limitations in mind. First, study results are based on weighted data, which should reflect the distributions in the population of noninstitutionalized, English-speaking U.S. adults aged 18 and older; that said, the NORC survey weights incorporate several variables (gender, age, education, race/ethnicity, and region) but not political affiliation. There are more Democrats in the study sample, and generalizability might be constrained with respect to political affiliation. Second, survey space constraints limited our ability to look at disagreement *between* health experts and politicians. Our global assessment of cross-source exposure to conflicting information surrounding COVID-19 suggests that perceptions of such disagreement likely exist, but future research should directly address this question. Third, to avoid priming conflict and activating responses, the source- and issue-specific measures of perceived disagreement preceded the global assessment measure; however, this ordering may have resulted in overreporting of cross-source exposure to conflicting information about COVID-19. Fourth, although these measures were informed by past research [27, 36], they did not undergo cognitive testing prior to study launch; lower literacy participants might have found the definitions to be challenging. Fifth, although this cross-sectional survey study enables us to identify factors that are associated with perceptions of disagreement, inferences regarding causality cannot be established. Sixth, we report perceptions observed in the U.S.; it is not clear whether similar perceptions have emerged in other countries, though some research hints at this possibility [43]. Last, the four aspects and six strategies were selected from the universe of COVID-19-related issues prevalent in the media during mid/late April; given the rapid changes in COVID-19 science and, in turn, the fluidity of media and public discourse, it is conceivable that repeating this survey later in 2020 would yield different public perceptions of disagreement about one or more issues. It is for this reason that we report and predict the summary indices of the broader phenomena that those responses instantiate, rather than the issue-specific perceptions.

Rapidly evolving science necessarily can create conditions ripe for disagreement among experts, and when that science is politicized, politicians can be an additional source of conflict. This study suggests that, overall, such disagreements are not confined to professional circles, but instead they play out in the broader public information environment, and this content does not go unnoticed by the public. Whether such perceptions of disagreement are ultimately consequential is an empirical question that should be addressed in future longitudinal survey or experimental research—after all, just because someone perceives disagreement among politicians does not necessarily mean they are insecure in their own beliefs—but existing theory and research on the effects of exposure to conflicting health information raise cause for concern [8–11, 43]. Although researchers are well prepared to make sense of evolving scientific evidence and health recommendations, the public may struggle to do so, due in part to limited

literacy about scientific research [11]. Several experimental studies have found that exposure to conflicting health information produces negative emotional reactions to such content (such as frustration, annoyance, and distress) [9] and undesirable cognitive outcomes, including confusion (perceived ambiguity about the health topic in question or health research in general) [8–10], backlash (negative beliefs or attitudes toward the health topic in question or health research in general) [10, 44], and attitudinal ambivalence (positive and negative evaluations of a given object at the same time) [9, 44]. Moreover, there is some evidence that these affective and cognitive responses could translate into behavioral effects [8]. This is a pressing concern with COVID-19, as perceptions of conflict and disagreement could produce not only confusion about and decreased trust in recommendations but also reduced compliance with mitigation behaviors, including those for which there is substantial consensus (e.g., hand washing). Such undesirable effects could be particularly likely if those recommendations are perceived to be coming from politicians, given that, across issues, participants perceived politicians to be debating and disagreeing more than health experts. That said, our results suggest that such effects could vary across population subgroups and might be shaped by factors such as political affiliation, information source use, and personal experience with COVID-19.

This study's findings have important implications for public health communication research and practice. First, results underscore that the public perceives disagreement among health experts and, to an even greater degree, among politicians. Such patterns might not be confined to the COVID-19 context, but could in fact be likely whenever a health issue becomes politicized—a troubling trend that has been documented in recent years [33]. Because the source of conflicting information could influence both whether people notice it and how they respond to it, future research should examine the myriad sources of such content. Second, we need to anticipate that many people likely interpret local, state, and federal health department and organization COVID-19 recommendations against a backdrop of seemingly ever-shifting advice, which could undermine the effectiveness of that strategic messaging. Crisis risk communication research suggests that public health messengers should anticipate such perceived conflict and respond accordingly, by exhibiting compassion and explicitly acknowledging uncertainty and shifts [45]. Some efforts already have been undertaken here; for example, Harvard T.H. Chan School of Public Health, in its "COVID-19 Path Forward" plan, underscores unified guidance and consensus while simultaneously acknowledging and normalizing evolving scientific advice [46]. Health journalists could follow similar strategies in their reporting. Just as there are growing efforts to intervene and address COVID-19-related misinformation [3, 47, 48], so, too, should attention be devoted to addressing conflicting information.

## Supporting information

**S1 Table. Issue-specific perceptions of disagreement among health experts and politicians about aspects of COVID-19 (coronavirus).**
(DOCX)

**S2 Table. Issue-specific perceptions of disagreement among health experts and politicians about the effectiveness of strategies for preventing the spread of COVID-19 (coronavirus).**
(DOCX)

## Acknowledgments

The authors thank the participants for their time and interest in participating in this study. We also thank Michael Schommer and Kate Awsumb from the Minnesota Department of Health for their feedback and support.

## Author Contributions

**Conceptualization:** Rebekah H. Nagler, Sarah E. Gollust, Alexander J. Rothman, Erika Franklin Fowler, Marco C. Yzer.

**Formal analysis:** Rebekah H. Nagler, Rachel I. Vogel.

**Funding acquisition:** Rebekah H. Nagler.

**Methodology:** Rebekah H. Nagler, Rachel I. Vogel, Sarah E. Gollust.

**Project administration:** Rebekah H. Nagler.

**Resources:** Marco C. Yzer.

**Supervision:** Rebekah H. Nagler.

**Writing – original draft:** Rebekah H. Nagler.

**Writing – review & editing:** Rebekah H. Nagler, Rachel I. Vogel, Sarah E. Gollust, Alexander J. Rothman, Erika Franklin Fowler, Marco C. Yzer.

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
