## [Decision Letter · Decision Letter 0]

7 Aug 2020

PONE-D-20-22073

Public perceptions of conflicting information surrounding COVID-19: Results from a nationally representative survey of U.S. adults

PLOS ONE

Dear Dr. Nagler,

Thank you for submitting your manuscript to PLOS ONE. After careful consideration, we feel that it has merit but does not fully meet PLOS ONE’s publication criteria as it currently stands. Therefore, we invite you to submit a revised version of the manuscript that addresses the points raised during the review process.

We look forward to receiving your revised manuscript.

Kind regards,

Sze Yan Liu, PhD

Academic Editor

PLOS ONE

Journal Requirements:

2. Please improving statistical reporting and refer to p-values as "p<.001" instead of "p=.000". Our statistical reporting guidelines are available at https://journals.plos.org/plosone/s/submission-guidelines#loc-statistical-reporting

Reviewers' comments:

Reviewer's Responses to Questions

**Comments to the Author**

1. Is the manuscript technically sound, and do the data support the conclusions?

Reviewer #1: Yes

Reviewer #2: Yes

Reviewer #3: Yes

2. Has the statistical analysis been performed appropriately and rigorously? 

Reviewer #1: Yes

Reviewer #2: Yes

Reviewer #3: Yes

3. Have the authors made all data underlying the findings in their manuscript fully available?

Reviewer #1: Yes

Reviewer #2: Yes

Reviewer #3: Yes

4. Is the manuscript presented in an intelligible fashion and written in standard English?

Reviewer #1: Yes

Reviewer #2: Yes

Reviewer #3: Yes

5. Review Comments to the Author

Reviewer #1: Thank you for the opportunity to review “Public perceptions of conflicting information surrounding COVID-19: results from a nationally representative survey of U.S. adults.” The article presents interesting and timely findings regarding whether the public notices conflicting COVID information, particularly between health experts and politicians, and correlates of perception of conflicting information.

Abstract

• Well-written

Introduction

• The introduction is well-written and presents relevant previous studies and a rationale for the current study.

Methods

• Define the acronym NORC

• Recommend cutting the sentence on line 155 starting with, “as a multi-client.” Sounds like an unnecessary endorsement of NORC.

• Can you note how long the total survey was so we can get an idea of how long the survey was and respondent burden?

• Some of the conflicting information questions seem to be at a high level of literacy. For example, the definition of health experts is long and it is unlikely the general public knows what an “academic research institution” is. Were these questions piloted with individuals with low health literacy? If not, this should be noted as a limitation.

• Why was social media left off as a source for COVID-19 info?

• Line 277: please provide more detail about the regressions, including what the dependent variables were in the models.

• Please state what level of significance was (alpha)

Results

• Line 304: since the means are between 2 and 3, can you present those response options rather than the 1 and 4 response options. Can you also comment on whether these data were skewed or kurtotic?

• Line 311: please change p=.000 to p<0.001

• There are a lot of variables included in the regression- did you check for collinearity?

• Have you thought about entering the number of sources as a variable in the model rather than listing cable news, national news, local news, etc, separately?

Discussion

• Nice discussion of motivated reasoning

• Limitations should also mention that there were more Democrats and well-educated respondents in the sample and how this may affect the results

Reviewer #2: This is a very well written article dealing with a delicate topic. The paper presents the data of nationally representative surveys and study design and statistical analyses are strong. Thus, it appeal to readers' interest and/or help to understand the impact of conflicting information surrounding COVID-19 abounds on non-pharmaceutical intervention. Thanks for your outstanding contributions.

Reviewer #3: The purpose of this study was to examine perceptions of conflicting information from both health experts and politicians about COVID-19 among a nationally representative sample of slightly over 1,000 U.S. adults. Data were collected in late April 2020. Three-quarters of respondents reported being exposed to conflicting information, and the authors report demographic and other characteristics associated with perceptions of conflicting information. The study is primarily descriptive and is not hypothesis-driven. I enjoyed reading this paper and found the information to have great implications for health communication strategies surrounding COVID-19. The introduction does a very nice job of situating this study in the existing literature on conflicting health information. Strengths of the paper include a thorough and succinct introduction, national sampling procedures, counter balancing of measures (i.e. when measures were duplicated about health experts and politicians), and a strong discussion section. The methods and analyses are largely solid. As the authors note in the abstract, it will be important to understand how these beliefs are associated with cognitive and behavioral outcomes, and it is unfortunate that measures of that type were not included in this paper. I describe minor suggestions for improving the paper.

Introduction

1. Do the findings of this study have implications beyond understanding COVID? This could be addressed more in the intro and/or discussion.

2. Page 5, line 97: “what seem like daily swings in recommendations” sounds subjective and I think different terminology might be used to emphasize the frequency without this language.

3. I had difficulty understanding the definition of the politicization of health issues (pg 6, lines 122-123). An example might help to clarify.

Materials and Methods

4. For transparency, could the authors describe the other measures that were included in the survey but that are not analyzed here?

5. More of a description/explanation about NORC would be helpful. For example, what exactly does it mean that the panel “provides sample coverage of approximately 97% of the U.S. household population”? Could this be stated in more plain language? Also, how big is the panel, and do panel members complete multiple surveys?

6. Four summary indices of perceptions of conflicting information are reported. What is the justification for keeping these separate rather than combining all the items referring to health experts and all of those referring to politicians (for a total of two scales)? I would like to see the alphas of the scales and correlations among the scales.

7. Additional justification is needed for how the specific news sources were collapsed into the categories of cable news, national news, etc. I would think that the sources categorized as cable news tend to be more conservative than those classified as national news, which is a potential limitation/confound of the categories reported. I am not sure how meaningful the categories are as reported. I would imagine that relying on different news sources differs by political affiliation in meaningful ways, and the authors might consider accounting for this in their analyses.

Discussion

8. I was unclear what the sentence starting with “By comparison…” (page 23, lines 392-395) meant.

9. Please add citations to the statement that theory and research on the effects of exposure to conflicting health information raise cause for concern (page 25, lines 439-440).

6. PLOS authors have the option to publish the peer review history of their article (what does this mean?). If published, this will include your full peer review and any attached files.

Reviewer #1: No

Reviewer #2: No

Reviewer #3: No

---

## [Author Response · Author response to Decision Letter 0]

28 Aug 2020

RESPONSE TO REVIEWERS

MANUSCRIPT: Public perceptions of conflicting information surrounding COVID-19: Results 

from a nationally representative survey of U.S. adults (PONE-D-20-22073)

DATE: August 28, 2020 

Thank you very much for the expeditious review of our manuscript, “Public perceptions of conflicting information surrounding COVID-19: Results from a nationally representative survey of U.S. adults.” Based on the reviewers’ helpful suggestions, we have made several improvements to the manuscript. Below we outline the concerns raised and our responses. In the revised manuscript, we indicate all revisions using track changes.

Reviewer #1

1) In the sample and procedure section, define the acronym NORC; consider cutting the sentence on line 155, starting with “As a multi-client shared cost survey,” which sounds like an unnecessary endorsement of NORC; and note the total length of the survey.

We have added information on the total survey length to the sample and procedure section (pp. 7-8). We appreciate the reviewer’s point about the sentence on line 155. We agree that we should be mindful not to inadvertently suggest a dependent relationship, so we deleted the part that sounded like an endorsement. We do think it is important for readers to know that these data come from a multi-client shared cost survey so they can appreciate that the survey included questions that were not designed by our team (and thus are not part of the current project). For this reason, we retained that content. And strange though it may seem, NORC is not an acronym; rather, it is the organization’s formal name, “as with organizations such as IBM, AT&T, RAND, and GEICO” (https://www.norc.org/about/Pages/about-our-name.aspx).

2) Some of the conflicting information questions seem to be at a high level of literacy. For example, the definition of health experts is long, and it is unlikely the general public knows what an “academic research institution” is. Were these questions piloted with individuals with low health literacy? If not, this should be noted as a limitation.

The reviewer raises a good point. Although our measurement approach was informed by past 

research on self-reported exposure to conflicting health information and perceptions of politicized

health controversies, the specific measures used in this study were not piloted prior to study launch 

due to time constraints (i.e., we prioritized timely data collection given the rapidly evolving crisis). 

In developing our health experts definition, we did look to public opinion surveys that asked about 

trust in specific health expert sources (e.g., https://www.kff.org/coronavirus-covid-19/report/kff-health-tracking-poll-early-april-2020/), which might allay some concerns about differential understanding across literacy levels. We nonetheless now note this as a limitation (p. 26). 

3) Why was social media left off as a source for COVID-19 information?

This survey question, which asked participants about the “specific information sources” they turned to for information about COVID-19, focused on substantive source content rather than the precise vehicle of information delivery. For example, if a participant selected “NPR or its website,” “National network news or their websites,” and “World Health Organization (WHO),” that tells us something about the types of substantive information sources they sought out, but in each case they might have engaged with that content via a different vehicle. In other words, they might have listened to NPR on the radio, visited the NPR website, and/or followed NPR on Twitter. Regardless of the vehicle of delivery, the type of content (which, in the case of NPR, would be national news content) is the same. Although it is true that there could be some variation in information engagement patterns depending on the vehicle of information delivery (e.g., a participant might engage more deeply with NPR content they seek out and read online versus hear in passing on the car radio or in a Tweet they see on a timeline), survey space constraints precluded this more nuanced parsing.

4) In the analytic approach section, please provide more detail about the regressions, including what the dependent variables were in the models, and state what the significance level was (alpha). Also, did you check for collinearity? There are a lot of variables in the regression models.

We added more details about the regressions and stated the significance level; we also specify that there was no evidence of collinearity in the regression models (pp. 13-14). Most Spearman correlation coefficients among independent variables were less than 0.1, and none were greater than 0.3; additionally, most variance inflation factors were below 3, and all were well below 10.

5) In the results section, since the means for the summary indices are between 2 and 3, present those response options instead (line 304), and change the p-value on line 311 to <0.001. For the summary indices, can you comment on whether those data were skewed or kurtotic? 

These changes have been made (pp. 16-17). Regarding the question of skewness and kurtosis, the indices do not deviate substantially from normality (statistics are listed below). The aspects indices are symmetric, though lighter tailed; the strategies indices are moderately to highly skewed, and the health experts distribution is heavier tailed. 

Aspects index, health experts: skewness = 0.1, kurtosis = -0.9

Aspects index, politicians: skewness = -0.4, kurtosis = -0.5

Strategies index, health experts: skewness = 1.5, kurtosis = 2.4

Strategies index, politicians: skewness = 0.7, kurtosis = 0.2

6) Have you thought about entering the number of sources as a variable in the model, rather than listing cable news, national news, local news, etc. separately?

We appreciate the reviewer’s question, as we agree that the volume of information exposure also could prove important. To consider this possibility, we asked participants to report the frequency with which they actively checked for news about COVID-19 from any source (p. 13). Although tallying the number of information sources could be an alternative approach to assessing the volume of exposure, conceivably someone might report using only two sources but actually use them quite heavily, while someone else might report using five or more sources but use them sporadically at best. We believe the self-reported assessment of frequency of checking for news provides a useful complement to the source-specific measures of information use.

7) Mention as a limitation that there were more Democrats and well-educated respondents in the sample, noting how this might affect the results.

We thank the reviewer for raising this issue. Study results are based on weighted data, which should reflect distributions in the population. This allays many, though not all, of the concerns about generalizability. We now mention this in the limitations section (p. 25).

Reviewer #2

This is a very well written article dealing with a delicate topic. The paper presents data from a nationally representative survey, and study design and statistical analyses are strong. Thus, it appeals to readers' interest and/or helps to understand the impact of conflicting information surrounding COVID-19. Thanks for your outstanding contributions.

Thank you very much; we appreciate your taking the time to review this work.

Reviewer #3

1) Do the findings of this study have implications beyond understanding COVID? This could be addressed more in the introduction and/or discussion.

We thank the reviewer for asking this question, as we do believe our findings have implications beyond the COVID-19 context. We therefore added text to the implications section of the discussion (pp. 27-28), where we describe the importance of the public perceiving disagreement not only among health experts but also (and to an even greater degree) among politicians. We note how such source patterns might be likely whenever a health issue becomes politicized.

2) On line 97, “what seem like daily swings in recommendations” sounds subjective; different terminology might be used to emphasize the frequency.

We have reworded this clause to read: “what the public might perceive to be frequent shifts in recommendations” (p. 5).

3) On lines 122-123, it was difficult to understand the definition of the politicization of health issues; consider adding an example to help clarify.

We have added an example to help clarify. The text now reads: “The politicization of health issues, defined as when political cues become integrated into those issues’ public presentation (e.g., when politicians’ perspectives appear in news media coverage of an issue to either endorse or highlight political conflict), has been well documented in recent years—among issues as wide-ranging as the human papillomavirus (HPV) vaccine, mammography screening, and the Affordable Care Act (ACA)) [33, 34]” (p. 6).

4) For transparency, describe the other measures that were included in the survey that are not analyzed here.

This information has been added on p. 8: “Data not analyzed here come from questions that assessed participants’ perceptions of disparities in COVID-19 mortality, other COVID-19-related cognitions (e.g., self-efficacy to reduce risk of infection), patterns of information avoidance, and past mitigation behaviors (e.g., stockpiling groceries and other supplies).” Some of these data are reported in a separate manuscript. If the editorial team prefers it, we would be glad to include in an appendix the full set of survey questions that we added to NORC’s AmeriSpeak Omnibus instrument. 

5) More of a description/explanation about NORC would be helpful. For example, what exactly does it mean that the panel “provides sample coverage of approximately 97% of the U.S. household population”; how big is the panel; and do panel members complete multiple surveys?

We have included additional information about NORC in the sample and procedure section (pp. 7-8). Specifically, we have clarified which types of households are excluded from the panel, noted the panel’s current size, and described panelists’ typical survey completion patterns.

6) Four summary indices of perceptions of conflicting information are reported. What is the justification for keeping these separate rather than combining all the items referring to health experts and all of those referring to politicians (for a total of two scales)? I would like to see the alphas of the scales and correlations among the scales.

For both the health expert and politician question blocks, we kept the indices separate because of the conceptual distinctions between them. All six items in the strategies index assess the effectiveness of strategies for preventing the spread of COVID-19. In contrast, the four items in the aspects index, though more loosely connected to one another, do not focus on the effectiveness of prevention strategies; rather, they are concerned with perceptions of risk, testing, and treatment. 

As requested, we are providing the alphas and correlations below; the alphas also have been added to the manuscript on p. 16. The aspects and strategies indices are not so strongly correlated so as to suggest they are capturing the same phenomena. 

Aspects index, health experts: Cronbach’s alpha = 0.79

Aspects index, politicians: Cronbach’s alpha = 0.80

Strategies index, health experts: Cronbach’s alpha = 0.84

Strategies index, politicians: Cronbach’s alpha = 0.85

Correlation between aspects and strategies indices, health experts = 0.55

Correlation between aspects and strategies indices, politicians = 0.55

7) Provide additional justification for how the specific news sources were collapsed into the categories of cable news, national news, etc. I would think that the sources categorized as cable news tend to be more conservative than those classified as national news, which is a potential limitation/confound of the categories reported. I also would imagine that relying on different news sources differs by political affiliation in meaningful ways; the authors might consider accounting for this in their analyses.

We appreciate the reviewer’s point and agree that we should have provided such justification. Our measurement decisions were informed by the Pew Research Center’s public opinion work on news and information sources, both in the COVID-19 context and more generally (see, for example, https://www.journalism.org/2020/03/25/americans-who-primarily-get-news-through-social-media-are-least-likely-to-follow-covid-19-coverage-most-likely-to-report-seeing-made-up-news/ and https://www.pewresearch.org/fact-tank/2018/01/05/fewer-americans-rely-on-tv-news-what-type-they-watch-varies-by-who-they-are/). We have added this information to the text (p. 12), as well as to the reference list.

As these examples illustrate, categorization in terms of local, national/network, and cable news is typical, with ideological variation within each category—perhaps particularly in the case of cable news (see, for example, https://www.journalism.org/2020/04/01/cable-tv-and-covid-19-how-americans-perceive-the-outbreak-and-view-media-coverage-differ-by-main-news-source/). Similarly, in our categorization, cable news encompasses more conservative (Fox News), more liberal (MSNBC), and more centrist (CNN) sources. Given this distribution, we would not say that the cable news category would necessarily be more conservative than those sources classified as national news. 

We agree that relying on different news sources may vary by political affiliation, particularly for certain types of sources. We had considered including source by affiliation interaction terms in our models, but decided against this approach for two reasons. First, this was not an a priori research question (i.e., we had no clear rationale to believe that these factors, in combination, would be associated with varying perceptions of disagreement). Second, as Reviewer 1 notes, many independent variables are already included in the models, given our research question on correlates; thus probing interaction terms, especially without a priori justification, would likely be problematic from a collinearity and power perspective. Moreover, one might imagine that other interactions (e.g., with age or education) would be worth investigation; absent clear subgroup-related research questions or hypotheses at the outset, it would have been hard to know where to draw the line. Ultimately, we agree that future research should consider the interplay between specific sources and political affiliation (e.g., as has been conducted in the context of COVID-19 misinformation and right-leaning media; https://www.ncbi.nlm.nih.gov/pmc/articles/PMC7251254/). 

8) In the discussion, clarify the sentence that starts with “By comparison…” (on lines 392-395); and add citations to the statement that theory and research on the effects of exposure to conflicting health information raises cause for concern (on lines 439-440).

We have made these changes on p. 24 and p. 27.

Thank you again for the opportunity to revise and improve this manuscript. We are hopeful that this paper is now acceptable for publication in PLOS ONE.

---

## [Decision Letter · Decision Letter 1]

29 Sep 2020

PONE-D-20-22073R1

Public perceptions of conflicting information surrounding COVID-19: Results from a nationally representative survey of U.S. adults

PLOS ONE

Dear Dr. Nagler,

Thank you for submitting your manuscript to PLOS ONE. After careful consideration, we feel that it has merit but does not fully meet PLOS ONE’s publication criteria as it currently stands. Therefore, we invite you to submit a revised version of the manuscript that addresses the points raised during the review process.

As noted by the reviewers' the authors did an excellent job responding to earlier concerns.  There are two minor suggestions that we would like to authors to add to the manuscript to further strengthen it.

We look forward to receiving your revised manuscript.

Kind regards,

Sze Yan Liu, PhD

Academic Editor

PLOS ONE

Reviewers' comments:

Reviewer's Responses to Questions

**Comments to the Author**

1. If the authors have adequately addressed your comments raised in a previous round of review and you feel that this manuscript is now acceptable for publication, you may indicate that here to bypass the “Comments to the Author” section, enter your conflict of interest statement in the “Confidential to Editor” section, and submit your "Accept" recommendation.

Reviewer #1: All comments have been addressed

Reviewer #3: (No Response)

2. Is the manuscript technically sound, and do the data support the conclusions?

Reviewer #1: Yes

Reviewer #3: Yes

3. Has the statistical analysis been performed appropriately and rigorously? 

Reviewer #1: Yes

Reviewer #3: Yes

4. Have the authors made all data underlying the findings in their manuscript fully available?

Reviewer #1: Yes

Reviewer #3: (No Response)

5. Is the manuscript presented in an intelligible fashion and written in standard English?

Reviewer #1: Yes

Reviewer #3: Yes

6. Review Comments to the Author

Reviewer #1: The authors did an excellent job responding to reviewer comments. I look forward to seeing this published.

Reviewer #3: The authors were very responsive to the reviews and have improved the manuscript. I just have two minor additional comments related to issues that were addressed in the response to the reviewers but that I believe could be addressed more directly in the manuscript.

First, the authors state that some of the data not reported here are reported in a separate manuscript (Reviewer #3, Point 4). For transparency, I think the authors should cite the other paper (even if still in preparation or under review) and briefly describe the content so that reviewers/readers can be confident that the papers do not overlap.

Second, the authors describe the conceptual distinction between aspects and strategies related to COVID-19 in the letter (Reviewer #3, Point 6), but I did not see this described in the manuscript. I would recommend adding the justification from the letter to the manuscript, and perhaps including the correlations between the scales as well, in order to justify this decision to the reader.

7. PLOS authors have the option to publish the peer review history of their article (what does this mean?). If published, this will include your full peer review and any attached files.

Reviewer #1: No

Reviewer #3: No

---

## [Author Response · Author response to Decision Letter 1]

1 Oct 2020

RESPONSE TO REVIEWERS

MANUSCRIPT: Public perceptions of conflicting information surrounding COVID-19: Results from a nationally representative survey of U.S. adults (PONE-D-20-22073R1)

DATE: September 29, 2020 

Thank you very much for the review of our manuscript, “Public perceptions of conflicting information surrounding COVID-19: Results from a nationally representative survey of U.S. adults.” Based on the reviewers’ helpful suggestions, we have made two additional improvements to the manuscript. Below we outline the concerns raised and our responses. In the revised manuscript, we indicate all revisions using track changes.

Reviewer #1

The authors did an excellent job responding to reviewer comments. I look forward to seeing this published.

Thank you very much; we appreciate your taking the time to review this work.

Reviewer #3

1) The authors were very responsive to the reviews and have improved the manuscript. I just have two minor additional comments related to issues that were addressed in the response to the reviewers but that I believe could be addressed more directly in the manuscript.

First, the authors state that some of the data not reported here are reported in a separate manuscript (Reviewer #3, Point 4). For transparency, I think the authors should cite the other paper (even if still in preparation or under review) and briefly describe the content so that reviewers/readers can be confident that the papers do not overlap.

We now cite this paper, which was recently accepted for publication, and the following text has been added to p. 8: “Data that describe levels of public awareness of disparities in COVID-19 mortality and correlates of that awareness are reported elsewhere.[36]”

2) Second, the authors describe the conceptual distinction between aspects and strategies related to COVID-19 in the letter (Reviewer #3, Point 6), but I did not see this described in the manuscript. I would recommend adding the justification from the letter to the manuscript, and perhaps including the correlations between the scales as well, in order to justify this decision to the reader.

We thank the reviewer for this suggestion. The following text has been added to pp. 10-11: “For both the health expert and politician question blocks, we kept the aspects and strategies indices separate because of the conceptual distinctions between them. All six items in the strategies index assess the effectiveness of strategies for preventing the spread of COVID-19; the four items in the aspects index, though more loosely connected to one another, are concerned with perceptions of risk, testing, and treatment. The indices are not so strongly correlated so as to suggest they are capturing the same phenomena (aspects and strategies indices, health experts: r = 0.55; politicians: r = 0.55).”

Thank you again for the opportunity to revise and improve this manuscript. We are hopeful that this paper is now acceptable for publication in PLOS ONE.

---

## [Editor Report · Decision Letter 2]

5 Oct 2020

Public perceptions of conflicting information surrounding COVID-19: Results from a nationally representative survey of U.S. adults

PONE-D-20-22073R2

Dear Dr. Nagler,

We’re pleased to inform you that your manuscript has been judged scientifically suitable for publication and will be formally accepted for publication once it meets all outstanding technical requirements.

Kind regards,

Sze Yan Liu, PhD

Academic Editor

PLOS ONE
---

## [Editor Report · Acceptance letter]

13 Oct 2020

PONE-D-20-22073R2 

Public perceptions of conflicting information surrounding COVID-19:Results from a nationally representative survey of U.S. adults 

Dear Dr. Nagler:

I'm pleased to inform you that your manuscript has been deemed suitable for publication in PLOS ONE. Congratulations! Your manuscript is now with our production department. 

Kind regards, 

on behalf of

Dr. Sze Yan Liu 

Academic Editor

PLOS ONE